# A Comparison of Lymphoid and Myeloid Cells Derived from Human Hematopoietic Stem Cells Xenografted into NOD-Derived Mouse Strains

**DOI:** 10.3390/microorganisms11061548

**Published:** 2023-06-10

**Authors:** Hernando Gutierrez-Barbosa, Sandra Medina-Moreno, Federico Perdomo-Celis, Harry Davis, Carolina Coronel-Ruiz, Juan C. Zapata, Joel V. Chua

**Affiliations:** 1Institute of Human Virology, University of Maryland School of Medicine, Baltimore, MD 21201, USA; smedinamoreno@gmail.com (S.M.-M.); hdavis@ihv.umaryland.edu (H.D.); jczapata2019@gmail.com (J.C.Z.); 2Instituto de Genética Humana, Facultad de Medicina, Pontificia Universidad Javeriana, Bogotá 110231, Colombia; perdomo_federico@javeriana.edu.co; 3Vice-Chancellor of Research, Virology Group, Universidad El Bosque, Bogotá 110121, Colombia; caritocruiz@hotmail.com

**Keywords:** humanization, humanized mouse model, xenograft, CD34, NSG, NCG, NOG-EXL, NSG-SGM3, lymphoid, myeloid

## Abstract

Humanized mice are an invaluable tool for investigating human diseases such as cancer, infectious diseases, and graft-versus-host disease (GvHD). However, it is crucial to understand the strengths and limitations of humanized mice and select the most appropriate model. In this study, we describe the development of the human lymphoid and myeloid lineages using a flow cytometric analysis in four humanized mouse models derived from NOD mice xenotransplanted with CD34^+^ fetal cord blood from a single donor. Our results showed that all murine strains sustained human immune cells within a proinflammatory environment induced by GvHD. However, the Hu-SGM3 model consistently generated higher numbers of human T cells, monocytes, dendritic cells, mast cells, and megakaryocytes, and a low number of circulating platelets showing an activated profile when compared with the other murine strains. The hu-NOG-EXL model had a similar cell development profile but a higher number of circulating platelets with an inactivated state, and the hu-NSG and hu-NCG developed low frequencies of immune cells compared with the other models. Interestingly, only the hu-SGM3 and hu-EXL models developed mast cells. In conclusion, our findings highlight the importance of selecting the appropriate humanized mouse model for specific research questions, considering the strengths and limitations of each model and the immune cell populations of interest.

## 1. Introduction

Humanized mice have been a crucial model for studying different components of the immune system, such as the interaction between infectious diseases and the human immune response, the development of cancer within a humanized immune response, and the rise of graft-versus-host disease (GvHD). However, the development of these models through human hematopoietic stem cell (HSC) xenoengraftment in immunodeficient mice faces two major challenges: preventing rejection and supporting human cell hematopoiesis. Xenotransplant rejection is caused by the classical complement pathway along with the humoral and cellular immune responses, involving key mouse immune cells such as macrophages, natural killers (NKs), T cells, B cells, and dendritic cells (DCs) [1]. Additionally, the lack of cross-reactivity between murine and human cytokines, as well as growth factors necessary for human myeloid and lymphoid cell development, causes defective engraftment of human immune cells. For instance, some of the most relevant cytokines involved in hematopoiesis, such as interleukin (IL)-3, stem cell factor (SCF), and granulocyte/macrophage colony-stimulating factor (GM-CSF), have less than 2.5-fold cross-reactivity between mouse and human cytokines and their receptors [2].

In the search to overcome xenograft rejection, different mouse models have been developed. One of the most commonly used strains is the NOD *scid* gamma or NSG (NOD-SCID-IL2null^−/−^), which combines three key modifications: the deficiency of the C5 complement protein that reduces the engraftment rejection mediated by the complement pathway; the SCID mutation that prevents TCR/BCR development in mouse T and B cells; and a mutation in the IL2 receptor (IL2R gamma null-chain receptor), required for the signaling of key cytokines such as IL-2, IL-3, IL-7, IL-9, IL-15, and IL-21, which are all involved in the development of murine T/B cells, NK cells, and dendritic cells (DCs) [3,4,5,6,7]. Another strain is the NCG, co-developed by the Nanjing Biomedical Research Institute of Nanjing University and Nanjing Galaxy Biopharma in 2014 and transferred to Charles River in 2016. The NCG mouse not only has the IL-2 gamma receptor deficiency but also a polymorphism in the *Sirpa alpha* (*SIRP a*) gene, reducing the phagocytic activity of murine macrophages when they recognize human cells [8].

In recent years, other immunodeficient mouse strains derived from the NOD strains have been developed with the goal of not only preventing xenotransplant rejection but also supporting and maintaining human cell hematopoiesis through the expression of human cytokines such as IL-3, GM-CSF, and SCF. These mouse strains include NSG-SGM3 and NOG-EXL (hGM-CSF/hIL-3 NOG) [9,10] which, in addition to the NSG genotype, also express human cytokines at different concentrations. As such, NOG-EXL mice express human IL-3 at concentrations of ~3–80 pg/mL and GM-CSF ~35 pg/mL, while NSG-SGM3 mice express IL-3 at ~2000–4000 pg/mL, GM-CSF, and KITLG (SF) ~2000–4000 pg/mL [9,11]. These differences in cytokine expression may affect the engraftment level and differentiation of hHSC cells and should be considered when choosing a mouse model for research.

Previous reports have demonstrated improved engraftment of hCD45^+^ cells, monocytes, and CD4^+^ T cells, as well as better development of human mast cells in NSG-SGM3 and NOG-EXL mice compared to NSG [10,12,13,14]. However, a direct comparison between different models designed to reduce xenograft rejection (NSG and NCG), and strains intended to sustain hematopoietic development (NOG-EXL and NSG-SGM3), have not yet been reported. This direct comparison is of relevance, considering that the level of engraftment and development of human cells can be influenced by the source of hematopoietic stem cells, the donor, and the reconstitution and preconditioning methods [15,16,17].

Therefore, this study aimed to compare the ability of four immunodeficient strains (NSG, NCG, NOG-EXL, and NSG-SGM3) to support and develop a human xenograft of CD34^+^ HSCs from fetal cord blood and describe the potential applications in human disease, including cancer, infectious disease, and graft-versus-host-disease (GvHD). All four strains were xenotransplanted using HSCs from a single donor derived from cord blood, allowing for a direct comparison among strains. Our results demonstrate that mouse strains that combine strategies for preventing xenotransplant rejection and express human cytokines (NSG-SGM3 and NOG-EXL) allow for the development of a wider pool of human immune cells.

## 2. Materials and Methods

### 2.1. Statement of Ethics

All animal care and procedures were performed according to protocols reviewed and approved by the Institutional Animal Care and Use Committee (IACUC) at the University of Maryland School of Medicine.

### 2.2. Mice and Xenotransplantation

NSG (NOD.Cg-*Prkdc^scid^ Il2rg^tm1Wjl^*/SzJ) and NSG-SGM3 (NOD.Cg-*Prkdc^scid^ Il2rg^tm1Wjl^* Tg (CMV-IL3,CSF2,KITLG)1Eav/MloySzJ) mice were purchased from Jackson Laboratories (n = 12 and n = 11). NCG (NOD-*Prkdc^em26Cd52^Il2rg^em26Cd22^*/NjuCrl) mice were purchased from Charles River Laboratories (n = 10), and NOG-EXL (NOD.Cg-*Prkdc^scid^ Il2rg^tm1Sug^* Tg (SV40/HTLV-IL3,CSF2)10-7Jic/JicTac) mice were purchased from Taconic Biosciences (n = 19). All mice were maintained at the Institute of Human Virology at the University of Maryland School of Medicine, Baltimore, USA, and housed in ventilated micro-isolation cages with autoclaved water and irradiated food in a high-barrier facility under specific pathogen-free conditions. For the xenotransplant, human cord blood from an anonymous donor was purchased from LONZA. Only one donor batch was used to reconstitute all four hu-mice strains.

NSG, NSG-SGM3, and NCG mice pups of 4 to 5 days old were irradiated four hours before engraftment with 1.1 cGy for 30 s from an X-ray irradiator. Thereafter, mice were intrahepatically injected with 50 μL of hCD34+ in PBS 1× with 80,000 cells. Since NOG-EXL mice are IL2rgtm1Sug heterozygous and need to be genotyped when they are born, 14–16-week-old mice were conditioned with 25 mg/kg of busulfan given 48 and 24 h intraperitoneally before a vein injection of 50 μL of hCD34^+^ with 80,000 cells.

### 2.3. Cytometric Bead Array (CBA) Analysis

Plasma samples at 14 weeks after xenotransplantation were collected and tested for human IL-6 (#558276 BD Biosciences), IFN-γ (#558269 BD Biosciences), IFN-α (#560379 BD Biosciences), TNF (#560112 BD Biosciences), IL12p70 (#558283 BD Biosciences), IL10 (#558274 BD Biosciences), and IP10 (#558280 BD Biosciences) using the CBA array (#558264 BD Biosciences). Standard curves using human proteins were used to measure cytokine concentration. Data were analyzed with FlowJo Version 10.

### 2.4. Flow Cytometry Assays

Fourteen weeks after CD34^+^ cell infusion, samples of peripheral blood, bone marrow, and spleen were harvested from recipient mice. Human lymphoid cells were analyzed based on the expression of hCD45 (Clone 2D1), hCD20 (2H7), hCD3 (UCHT1), hCD4 (OKT4), and hCD8 (SK1). Human myeloid cells were analyzed based on the expression of hCD45^+^ (2D1), hCD3^−^ (UCHT1), and hCD16+ (3G8) for monocytes. hCD45^+^ (2D1), HLA-DR^−^ (LN3), CD1c (L161), CD123 (6H6), Lin 1 (CD3 (UCHT1), CD14 (HCD14), CD16 (3G8), CD19 (HIB19), CD20 (2H7), CD56 (HCD56)) for dendritic cells. hCD45^+^ (2D1), HLA-DR^−^ (LN3), CD66b^−^ (G10F5), FcErI^+^ (CRA1), and CD117^+^ (104D2) for mast cells. hCD41^+^ (HIP8), hCD62P^+/−^ (AK4) for platelets. hCD41^+^ (HIP8), hCD42b^−^ (HIP1) immature megakaryocytes and hCD41^+^ (HIP8), and hCD42b^+^ (HIP1) for mature megakaryocytes. A representative gating strategy is shown in Appendix A.

### 2.5. Detection of Platelets

For platelet detection in the hu-mouse models, 30 μL of peripheral blood was collected in 600 μL of a megakaryocyte buffer (0.1 mM theophylline (Sigma, Burlington, NJ, USA), 15 mM sodium citrate (Sigma), and 1% BSA (Sigma) in PBS 1× sterile) [18], followed by centrifugation at 100× *g* for 15 min at room temperature (RT). Then, the supernatant was centrifuged at 1000× *g* for 10 min and subsequently discarded, and the pellet was collected. The platelets were cautiously resuspended in PBS 1× to avoid undesired activation and stained with those previously mentioned antibodies [19].

### 2.6. Bone Marrow Dissection

Bone marrow dissection was performed as previously described [20]. Briefly, mice were euthanized by CO_2_ asphyxiation, followed by cervical dislocation. Femurs were dissected from the mouse, excess tissue was removed, and the epiphyses were snipped off. Bone marrow was flushed with Iscove’s Modified Dulbecco’s Medium (IMDM) + 10% FBS (Gibco) using a 26.5-gauge needle and collected in a 50 mL falcon tube at room temperature, and then the solution was centrifuged at 300× *g* for 5 min. Then, cells were washed 2× with Hanks’ balanced salt solution (HBSS) (Gibco) and counted using a hemocytometer. Approximately 500,000 cells were stained using the antibody panels previously described.

### 2.7. Liver and Spleen Dissection

Mice were euthanized by CO_2_ asphyxiation in an acrylic plastic chamber for approximately 6–10 min, followed by cervical dislocation. Separately, the spleen and liver were dissected, placed in RPMI+ 10% FBS, passed through a cell strainer, and each was resuspended in RPMI+ 10% FBS. The suspensions were layered onto a Ficoll-Paque premium 1084, spun for 3 min at 3000× *g*, and peripheral blood mononuclear cells (PBMCs) were retrieved as an opalescent layer between the media and the Ficoll. Aliquots of approximately 500,000 cells were stained using the antibodies previously described.

### 2.8. Statistical Analysis

Data from the experiments were presented as the mean plus standard deviation (SD) and graphed using GraphPad Prism Version 8. To test the differences in the mean values of human cell population frequencies, all data were subjected to a normality test using a Shapiro–Wilk test, after which a nonparametric Kruskal–Wallis test followed by a Dunn test was used. A *p*-value <0.05 was considered statistically significant.

## 3. Results

### 3.1. A Higher Frequency of Human CD3+ Cells in the Peripheral Blood and Tissues in hu-SGM3 Mice

To induce the development of lymphoid linage in mice, human CD34^+^ cells were xenotransplanted into NSG, NCG, NOG-EXL, and NSG-SGM3 mice, and the frequencies of hCD45^+^, hCD3^+^, hCD20^+^, and hCD4:hCD8 ratio cells were initially analyzed at 14 weeks post-engraftment (Figure 1). The chimerism of hCD45^+^ cells in peripheral blood (PB) was significantly higher in frequency in hu-EXL mice (80%) than hu-SGM3 (60%), hu-NSG (55%), and hu-NCG (55%) (Figure 1A). For the lymphoid lineage, the hu-SGM3 mice had significantly higher frequencies of hCD3^+^, lower hCD20^+^, and a higher hCD4:hCD8 ratio, indicating that in this model, most of the T cell population was constituted by CD4^+^ T cells (Figure 1B–D).

In addition to peripheral blood (PB), we established the frequency of those cells in different organs at different time points (Figure 2, Figure 3 and Figure 4). The chimerism of human leucocytes was consistent between PB and tissues for each mouse strain. The hu-SGM3 model showed higher hCD3^+^ cells in both spleen and bone marrow (Figure 3B and Figure 4B) relative to the other mouse strains. Of note, the hu-NSG, hu-NCG, and hu-EXL models exhibited a stable xenoengraftment, with increased numbers of hCD8^+^ cells in PB and tissues. Overall, these data indicate that hu-SGM3 mice develop a greater lymphoid population with increased hCD3^+^ and lower hCD20^+^ compared with the other humanized mouse models.

### 3.2. Increased Frequency of Peripheral CD14^+^ Monocytes in the hu-SGM3 Model in the Blood and Tissues

Next, we assessed the impact of different mouse strains on the frequency of human CD14 (hCD14^+^) after engraftment (Figure 5). Even though all strains developed hCD14^+^ cells, the frequencies in PB were higher in hu-SGM3 in comparison to the other models. We investigated the frequency of this subset of cells in the spleen and bone marrow as well. The development of hCD14^+^ cells was markedly higher in the bone marrow in hu-SGM3 followed by hu-EXL, and lower percentages were found in the hu-NSG and hu-NCG models (Figure 5B). Although the percentage of hCD14^+^ cells was not statistically different in the spleens of all models, there were statistically higher frequencies of hCD14^+^ cells in hu-SGM3 peripheral blood and bone marrow (Figure 5C).

### 3.3. Enriched Frequency of Myeloid Dendricitc Cells in the Bone Marrow and Spleen in the hu-SGM3 Model

The frequencies of dendritic cells (DCs) in the peripheral blood and tissues were assessed in the different humanized mouse models (Figure 6 and Figure 7). While this population did not have high frequencies in the peripheral blood and only small differences were observed at weeks 14 and 15, hu-EXL showed an increase when compared to the other models (Figure 6A). The highest frequencies of DCs were observed in the bone marrow, followed by the liver and spleen in all the models (Figure 7A–C). Furthermore, in all the models, plasmacytoid dendritic cells (pDCs) constituted the major subset of DCs in tissues (Figure 7G–I). Interestingly, we found an enrichment of myeloid dendritic cells (mDC) in the hu-SGM3 model in the bone marrow and spleen (Figure 7D,E). These findings indicate that humanized mouse models can recapitulate a predominantly pDC immune response, and an enrichment of mDC in the hu-SGM3 model could be achieved in tissues.

### 3.4. Enhancement of Human Megakaryocyte Cell Development in the hu-SGM3 and hu-EXL Models

To assess the development and maturation profile of human megakaryocyte populations, we examined whether human cells expressing CD41a and CD42b were present. There were clear subsets of mature CD41a^+^ CD42b^+^ and immature CD41a^+^ CD42b^−^ cells in the bone marrow and spleen in all the humanized models (Figure 8). The frequencies of mature megakaryocytes were significantly higher in hu-SGM3 and hu-EXL in comparison to the other models (Figure 8A,C). Taken together, these data demonstrate that human megakaryopoiesis was more efficient in humanized mice expressing human cytokines.

### 3.5. Enhancement of Human Platelet Development in the hu-EXL Model

We determined the ability of the different humanized models to develop and maintain human platelets. The frequencies of human platelets CD41 (hCD41^+^) and their activation profiles were examined using hCD62 at different time points in PB (Figure 9). The presence of hCD41^+^ was significantly higher in hu-EXL than the other models throughout the analysis period with an inactivated profile of hCD41^+^hCD62^−^ (Figure 9A,B). Meanwhile, the frequency of hCD41^+^ cells did not differ statistically among hu-NSG, hu-NCG, and hu-SGM3. There was a tendency for lower circulating platelet frequency in the hu-SGM3 model, mainly with an activated platelet phenotype hCD41^+^hCD62^+^ (Figure 9B).

### 3.6. Enhancement of Human Mast Cell Development in the hu-SGM3 and hu-EXL Models

Considering the pathogenic roles of mast cells in allergic responses and some viral infections, such as DENV, we were interested in determining whether these cells can develop in these different humanized models. We examined the frequency of the phenotypic markers FcεRI and CD117 in the bone marrow and spleen (Figure 10). There was a statistically significant higher frequency of mast cells in the bone marrow and spleen samples from hu-SGM3 and hu-EXL mice than hu-NSG and hu-NCG (where the population of FcεRI^+^ CD117^+^ was not developed) (Figure 10A,B). This population was also tested in PB, and as expected, the frequency of FcεRI^+^ CD117^+^ was very low and was detected only in hu-SGM3 and hu-EXL. These results suggest that human cytokines expressed in hu-SGM3 and hu-EXL enhance the development of mast cells.

### 3.7. Human Cytokine Expression during Xenotransplantation in the hu-NSG, hu-NCG, hu-SGM3, and hu-EXL Humanized Mouse Models

To examine part of the inflammatory immune response in xenotransplant models, we tested seven inflammatory human cytokines (IL-6, IFN-γ, IFN-α, TNF-α, IL12p70, IL10, and IP10) in PB of the four humanized mouse models aforementioned. We detected low levels of IFN-γ in all models, and high and low levels of IL-6 in the hu-SGM3 and hu-EXL models, respectively (Figure 11, Table 1). These data demonstrate the capacity of the mature human cells to express human proinflammatory cytokines within the xenotransplant model and especially higher baseline levels of IL-6 in hu-SGM3 and hu-EXL mice.

## 4. Discussion

The hu-NSG models described here exhibited an efficient xenograft development with an initial high expansion of human B cells relative to T cells (CD4^+^ CD8^−^ and CD4^−^ CD8^+^) in the peripheral blood, as previously reported [10]. However, comparing successful engraftment between studies is challenging, as it is influenced by various factors such as mouse strain, age, source of CD34^+^ cells engrafted, preconditioning method, and donor variability [15,21,22,23]. It is also essential to consider the duration of experiments with humanized mouse models carefully. Although the strains described in this study do not develop thymic lymphoma over time, unlike the plain NOD/SCID strain [24], the maximum timeframe in the humanized mouse models is limited by the development of graft-versus-host disease, which will shorten the lifespan of the mice [23,25]. The impaired differentiation of human myeloid cells observed in humanized mouse models is likely related to the absence of specific signals during hematopoiesis. For instance, the lack of NK cells is associated with a deficit in key human cytokines, such as IL-15, which is crucial for the development of this cell population [26]. Our findings are consistent with this idea since none of the mouse models evaluated express human IL-15; thus, the NK population was not detected.

Similarly, macrophage colony-stimulating factor (M-CSF) plays an important role in the common myeloid progenitor (CMP) and promotes its differentiation into the monocyte/macrophage [27]. The absence of these human cytokines in hu-NSG and hu-NCG could explain the low proportion of monocytes found in those models, since studies have shown that expression of this human cytokine in humanized mice model enhances the development of the monocyte population [28].

The study of human professional antigen-presenting cells, known as dendritic cells from the myeloid and lymphoid lineages, faces several challenges. One of these is the low frequency of these cells in human blood, which makes it difficult to study. Additionally, studying DCs in human tissues can be challenging, and there are differences between murine and human DCs that must be considered [29,30,31]. To overcome some of these challenges, the mouse models described here can be used to study dendritic cells in tissues, particularly in the bone marrow. This tissue is identified as the compartment of the common DC progenitors [32].

All the humanized mouse models that we described showed an increase in plasmacytoid dendritic cells over the myeloid dendritic cells, making them suitable for studying immune responses that are predominately pDC-mediated [33]. The lower development of mDC could be due to the dependence on the ligand for the fms-like tyrosine kinase (Flt3) [34,35]. On the other side, an increase in the pDCs could be related to the deficit of the human arly hydrocarbon receptor (AhR) since antagonist-like StemRegnin 1 promotes human pDC development from HSC CD34^+^ in vitro [36], while in mice, the lack of AhR promotes murine pDC development [37]. Interestingly, we also found that the hu-SGM3 model showed a 2–6-fold increase in myeloid CD1c^+^ in the bone marrow and spleen compared to the hu-NSG and hu-NCG models, respectively. This finding is comparable to that described in the literature for the hu-SGM3 model and highlights the importance of human GM-CSF and IL3 in improving the development of human dendritic cells [10]. However, to enhance the antigen presentation function of the DC and the innate and cell-mediated immune response in the humanized mouse models, the expression of human HLA in mice is necessary, as described in other studies [38,39]. Overall, the hu-SGM3 and hu-EXL models, in combination with human HLA expression in the mice, will be most suitable for studying a wide range of infectious diseases and cancer.

The lack of mast cells, an important part of many inflammatory responses in the hu-NSG and hu-NCG models, can be partially attributed to insufficient cytokine supply, including G-CSF, GM-CSF, IL-3, IL-6, Fms-related tyrosine kinase 3 ligand, thrombopoietin (TPO), and stem cell factor (SC) [40,41,42,43,44]. Our results show that only the humanized mouse models with human cytokines (hu-SGM3 and hu-EXL) develop human mast cells. These data imply that the expression of cytokines, such as IL-3 and GM-CSF, in these models improves the development of human mast cell populations.

Overall, the hu-NSG mouse model has been extensively utilized for the study of infectious diseases, cancer, and parasitology (Table 2). This model is suitable for both long- and short-engraftment time experiments with a focus on T and B cells involved in the acquired or antigen-specific immune response. Additionally, this model can be used to study human megakaryocytes and platelets with additional modifications, such as the depletion of murine macrophages.

The NCG mouse model was selected for its macrophages expressing the transmembrane signal regulatory protein-α (SIRPA) with a polymorphism in the IgV-like domain. This allows for the interaction with the human ligand CD47, activating an inhibitory signal for phagocytosis by mouse macrophages, known as the “do not eat me” signal. The human CD47 is a member of the Ig superfamily expressed in hematopoietic and non-hematopoietic cells, and the interaction between the murine SIRPA with the polymorphism and the human CD47 has been associated with better xenografting [56]. However, in our study, we did not find any difference in the level of engraftment, lymphoid and myeloid lineages, and human platelets between the hu-NSG mice (SIRPA polymorphism negative) and the hu-NCG (SIRPA polymorphism positive). Even though the hu-NCG model has the SIRPA polymorphism, it still expresses a mouse SIRPA, which may affect the affinity with human CD47. To overcome this, the C57BL7G.Rag2^null^ILrg^null^ Sirpa ^human/human^ (BRGS^human^) with a knock-in allele human SIRPA demonstrates a high affinity for human CD47 mice as described [57]. This results in greater maintenance of human hematopoiesis with a high level of myeloid reconstitution. Overall, in this regard, the hu-NCG model has shown to have a similar advantage as the hu-NSG model and can be used as a model for studying HIV or other entities that require long-term engraftment (Table 2).

The limited biologic cross-reactivity among cytokines in humanized mice has been proposed as the mechanism for the misrepresentation of certain blood cell types [58]. Therefore, the use of immunodeficient mice, such as SGM3 and EXL, is important for the development of improved humanized mouse models. The hu-SGM3 model expresses IL-3, GM-CSF, and SCF human cytokines. In this model, we described an elevated number of human lymphoid and myeloid cells compared to similarly engrafted hu-NSG/NCG models. However, the most outstanding phenotype was the increase in T cells and lower amount of B cells in the peripheral blood, bone marrow, and spleen. The overexpansion of this cell population has been previously described; however, the mechanism is not derived by a direct action of the SCF, GM-CSF, or IL-3 since the cells committed toward the lymphoid lineage do not express the receptors for these cytokines. A possible mechanism could be derived from the effect of these cytokines on the progenitor cell or an indirect mechanism in the periphery rather than via enhanced thymic development [10].

Moreover, hu-SGM3 showed a higher number of monocytes and innate sensing/immune response cells in the spleen and bone marrow, which are the principal reservoirs of this population in mice [59,60], compared to the hu-EXL model. The increase in the number of monocytes is directly related to the higher levels of human IL-3 and GM-CSF expression in this model. Although these cytokines do not influence the late stages of monocyte maturation, they could enrich the early progenitor cells, indirectly impacting the final hCD14+ cells [12].

While the hu-SGM3 model produces a higher number of megakaryocytes a precursors of blood platelets, compared with the hu-EXL, hu-NSG, and hu-NCG models, it does not appear to be suitable for the study of platelet disorders, as evidenced by their low numbers and activated profile. Since the number of megakaryocytes is higher than in the other murine models, the reduced number of human circulating platelets may be due to platelet activation and consumption [61]. However, the exact mechanism of activation remains unknown in this system. Low platelet counts could also be related to the concentration of human cytokines expressed in the system since the hu-EXL model showed a comparable number of megakaryocytes with a higher number of human platelets in a resting stage.

Despite the absence of human cytokines in the hu-NSG and hu-NCG models, we found the presence of mature and immature megakaryocytes in both the bone marrow and spleen, which was consistent with previous reports [18]. However, it remains unclear how the mouse bone marrow microenvironment supports efficient megakaryocyte differentiation. This observation may imply complex interactions between mouse cytokines and human cytokines produced by other human cells developed in the bone marrow that may allow for the differentiation of megakaryocytes. Furthermore, the presence of this platelet-precursor population in hu-NSG mice suggests that the poor platelet reconstitution observed may result from the rapid clearance of human platelets by murine macrophages. This issue could be addressed by using clodronate liposomes to eliminate this population, thereby improving the presence of human platelets in the peripheral blood [62].

The stem cell factor plays a crucial role in mast cell survival, apoptosis inhibition [63], mast cell migration, adhesion, and IL-6 secretion [64,65,66]. Along with the SCF, interleukin-3 (IL-3) is another important cytokine that drives the development of mast cells. Although there have been contradictory studies about the role of IL-3 in human mast cell progenitor differentiation, recent studies demonstrate that it not only drives differentiation but also the survival of progenitor mast cells [67,68]. The presence of these cytokines in the hu-SGM3 and hu-EXL models allows for the development of mast cells, making these mouse models suitable for studying the function of this population during sepsis, allergy, and flavivirus infections [18,69,70,71,72]. Additionally, since we described an enrichment of mast cells in the bone marrow, these models could be used to study primary/secondary osteoporotic disorders, including rheumatoid arthritis and osteoarthritis pathologies that have an important component requiring mast cells [73,74].

The hu-SGM3 model is characterized by high numbers of T cells, with the presence of monocytes, megakaryocytes, and mast cells. However, this model has two principal limitations: the overactivation of human platelets and an early proinflammatory environment speeding up GvHD. The last limitation is shared with the hu-EXL model. This feature will impact the use of this model for long-term engraftment experiments as they could develop GvHD at a faster rate. Nevertheless, hu-SGM3 appears to be a good model for studying some viruses, such as HIV, and some cancers influenced by human cytokines, such as acute myeloma leukemia, and of course GvHD [33,75].

On the other hand, the hu-EXL mouse model is one of the best for developing human immune cell populations, including T/B cells, monocytes, megakaryocytes, resting human platelets, and mast cells. This feature makes it suitable for studying a wide variety of infectious diseases, such as HIV [33]. Additionally, one of the advantages of hu-EXL is the development of human platelets with a resting profile, making this model more suitable for the study of hemorrhagic fevers or megakaryocytic malignancies.

Finally, one important consideration when using humanized mouse models is the inflammatory environment produced by the xenografts during the development of GvHD. Initially, donor CD4+ and CD8+ cells recognize the host histocompatibility complex (MHC) on the antigen-presenting cells (APCs), and a mismatch drives the activation, proliferation, and differentiation of effector T cell subsets [76]. This is followed by the production and release of proinflammatory cytokines, such as IFN-γ and IL-17, produced by donor hematopoietic stem cell-derived Th1 and Th17, cells respectively [77]. IFN-γ makes host cells in the gastrointestinal tract and skin more susceptible to GvHD [78,79], while IL-17 will increase the production of IL-6 and IFN-γ in the early stage of the disease [80,81].

In the humanized mouse models described in this study, the expression of IFN-γ was found in all animals, but the expression of IL-6 was found primarily in hu-SGM3 and hu-EXL, the only two strains that developed mast cells. This profile could be due to the combination of an early GvHD with the enrichment of mast cell populations. Since this cell population can produce considerable amounts of IL-6 and increased numbers of mast cells are associated with increased GvHD [82,83], the proinflammatory background in the humanized mouse model must be considered when designing studies, especially if the inflammatory response is one of the aims of the study.

Several approaches have been developed in humanized mouse models to engraft human cells into immunodeficient mice. The optimal selection of the model requires a comprehensive understanding of the physiopathology of the disease and especially the capability of the model to allow the target cells to grow and be activated. Additionally, it is essential to have a broad understanding of the engraftment process and the overall strengths and limitations of the selected models. This work provides reference data on four humanized mouse models regarding the development and maintenance of different immune cell types, which can be used as a guide for an optimal humanized mice model selection to study specific human diseases.

## 5. Conclusions

The hu-NSG and hu-NCG models are excellent platforms to study human CD4^+^ T cells, CD8^+^ T Cells, CD14^+^ monocytes, dendritic cells, and megakaryocytes in different pathologies. On the other hand, humanized mouse models with the expression of human cytokines, such as hu-SGM3 and hu-EXL, not only enhance the development of human CD4^+^ T cells, CD8^+^ T cells, monocytes, dendritic cells, and megakaryocytes, but also allow the development of human mast cells. Additionally, hu-EXL develops better human platelet populations with a resting profile. However, there are some limitations in the humanized mouse models previously described that must be addressed. First, the different human cell profiles developed in each model; second, the low number of human platelets and their sensitivity to activation in the case of hu-SGM3; and finally, the proinflammatory microenvironment generated by the development of GvHD in all the models will decrease the lifespan of the mouse and create a proinflammatory background that must be considered when designing the study.

In conclusion, each immune-compromised mouse strain has particular characteristics that could be useful in the context of a specific research project. Therefore, researchers must carefully consider the strengths and limitations of the different humanized mouse models before selecting the most suitable one for their research proposes. Additionally, efforts must be made to address the limitations and challenges associated with the humanized models to enhance their efficacy as a tool for the study of various human diseases.

## Figures and Tables

**Figure 1 microorganisms-11-01548-f001:**
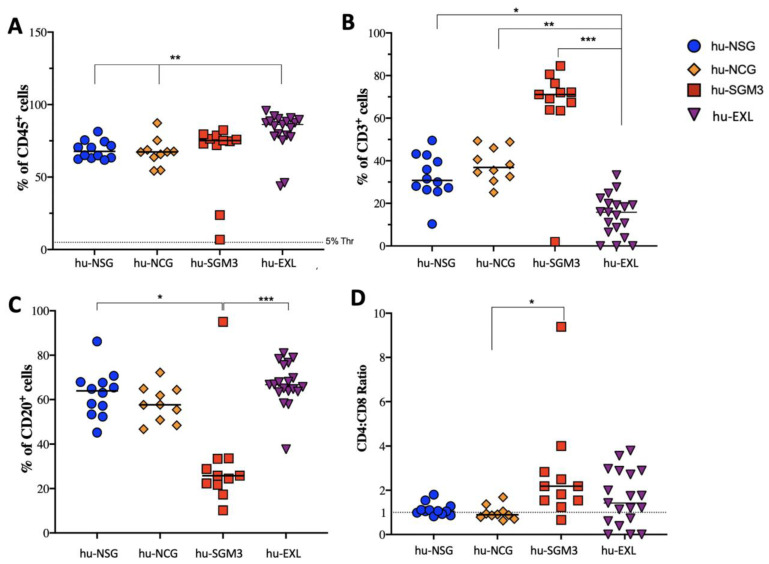
Frequency of lymphoid populations in the peripheral blood of the hu-NSG, hu-NCG, hu-SGM3, and hu-EXL models. The frequencies of hCD45^+^ (**A**), hCD3^+^ (**B**), and hCD20^+^ (**C**), and the hCD4:hCD8 ratio (**D**), were determined at 13 weeks post-engraftment and displayed as a mean value. Each group consisted of at least 10 mice. * *p* < 0.005; ** *p* < 0.01; *** *p* < 0.005 (Dunn test).

**Figure 2 microorganisms-11-01548-f002:**
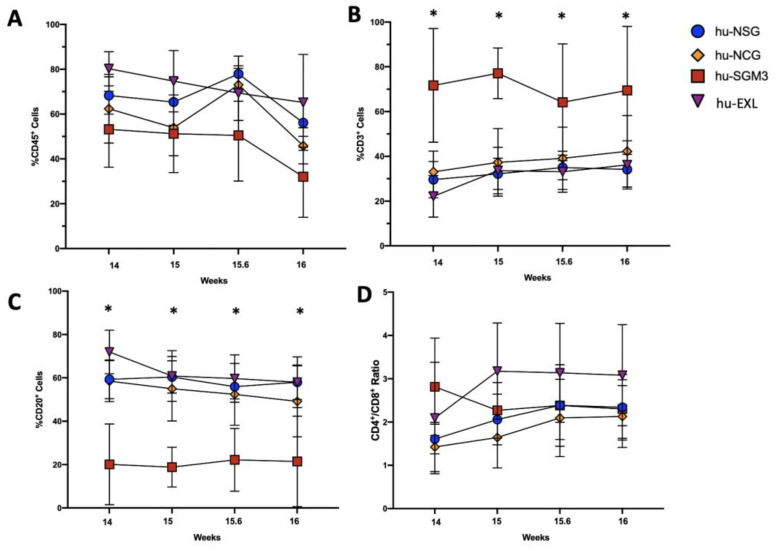
Frequency of lymphoid populations in the peripheral blood of the hu-NSG, hu-NCG, hu-SGM3, and hu-EXL models over time. Frequencies of human CD45^+^ (**A**), CD3^+^ (**B**), CD20^+^ (**C**), and the CD4:CD8 ratio (**D**) of all singlets. Each group contains at least 10 mice; horizontal bars indicate the mean value of each time point. * *p* < 0.005 (Dunn test).

**Figure 3 microorganisms-11-01548-f003:**
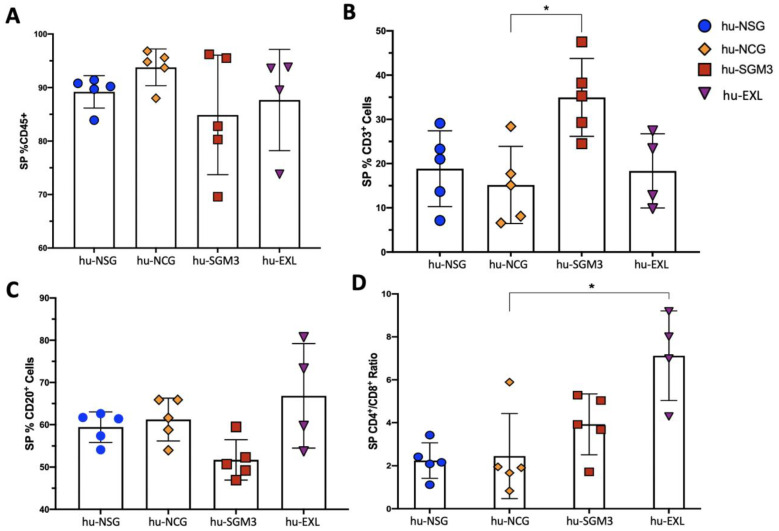
Frequency of the lymphoid populations in the spleen of the hu-NSG, hu-NCG, hu-SGM3, and hu-EXL models from weeks 15 and 16 combined. Frequencies of human CD45^+^ (**A**), CD3^+^ (**B**), CD20^+^ (**C**), and the CD4:CD8 ratio (**D**) of all singlets. Each group contains at least four mice; horizontal bars indicate the mean value. * *p* < 0.05 (Dunn test).

**Figure 4 microorganisms-11-01548-f004:**
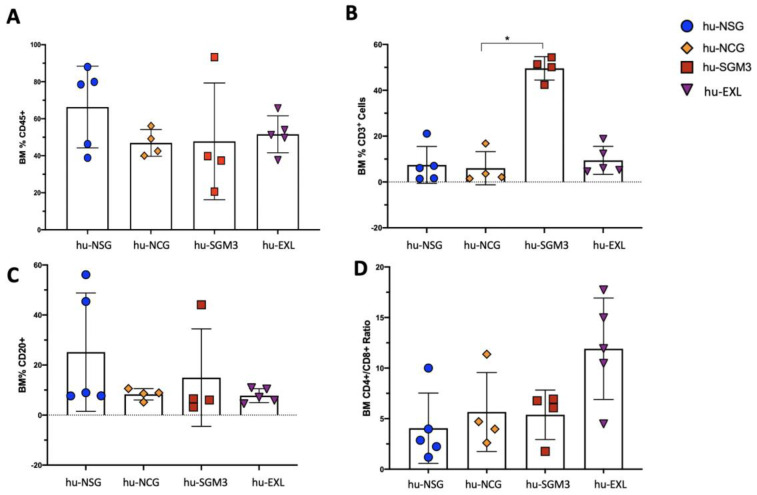
Frequency of lymphoid populations in the bone marrow of the hu-NSG, hu-NCG, hu-SGM3, and hu-EXL models from weeks 15 and 16 combined. Frequencies of human CD45^+^ (**A**), CD3^+^ (**B**), CD20^+^ (**C**), and the CD4:CD8 ratio (**D**) of all singlets. Each group contains at least four mice; horizontal bars indicate the mean value. * *p* < 0.05 (Dunn test).

**Figure 5 microorganisms-11-01548-f005:**
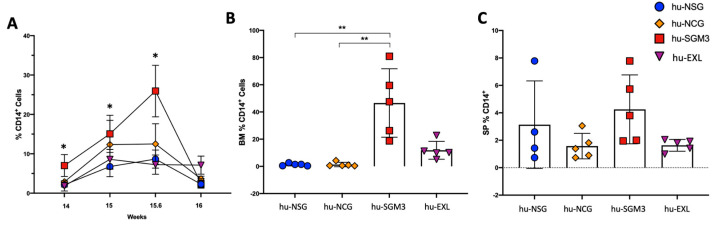
Frequency of monocytes in the peripheral blood and tissues of the hu-NSG, hu-NCG, hu-SGM3, and hu-EXL models. Frequencies of CD14^+^ in the peripheral blood (**A**), CD14^+^ in the bone marrow (**B**), and CD14^+^ in the spleen (**C**) from weeks 15 and 16 combined. Horizontal bars indicate the mean value. * Significant differences between hu-SGM3 and one or more humanized mouse models ** *p* < 0.001 (Dunn test).

**Figure 6 microorganisms-11-01548-f006:**
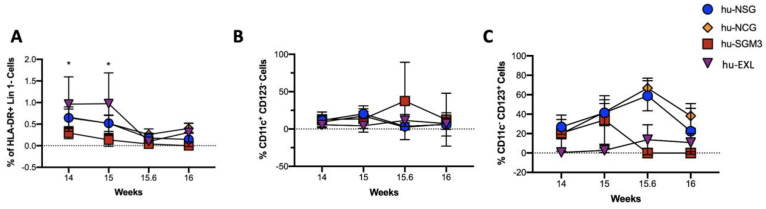
Frequency of dendritic cells in the peripheral blood of the hu-NSG, hu-NCG, hu-SGM3, and hu-EXL models. HLA-DR^+^ Lin 1^−^ cells from human CD45^+^ (**A**), CD1c^+^ CD123^−^ myeloid dendritic cells (mDCs) (**B**), and CD1c^−^ CD123^+^ plasmacytoid dendritic cells (pDCs) (**C**). Horizontal bars indicate the mean value. * *p* < 0.05 (Dunn test).

**Figure 7 microorganisms-11-01548-f007:**
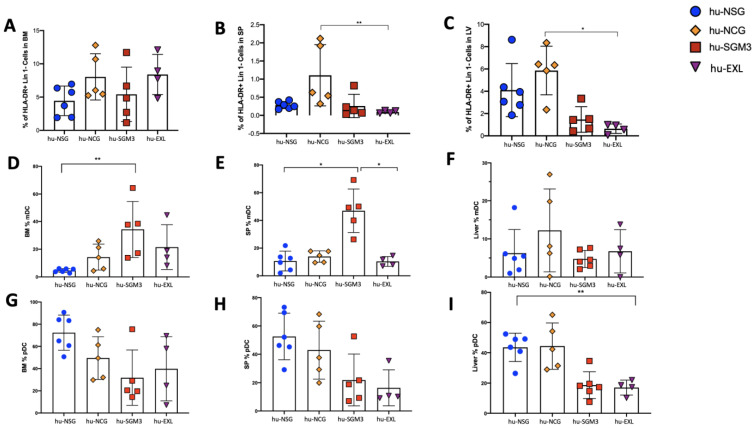
Frequency of dendritic cells in tissues of the hu-NSG, hu-NCG, hu-SGM3, and hu-EXL models from weeks 15 and 16 combined. Frequencies of HLA-DR^+^ Lin 1^−^ cells in the bone marrow (BM) (**A**), HLA-DR^+^ Lin 1^−^ cells in the spleen (SP) (**B**), HLA-DR^+^ Lin 1^−^ cells in the liver (LV) (**C**), CD1c^+^ CD123^−^ mDC in the bone marrow (**D**), CD1c^+^ CD123^−^ mDC in the spleen (**E**), CD1c^+^ CD123^−^ mDC in the liver (**F**), CD1c^−^ CD123^+^ pDC in the bone marrow (**G**), CD1c^−^ CD123^+^ pDC in the spleen (**H**), and CD1c^−^ CD123^+^ pDC in the liver (**I**). Horizontal bars indicate the mean value. * *p* < 0.05, ** *p* < 0.001 (Dunn test).

**Figure 8 microorganisms-11-01548-f008:**
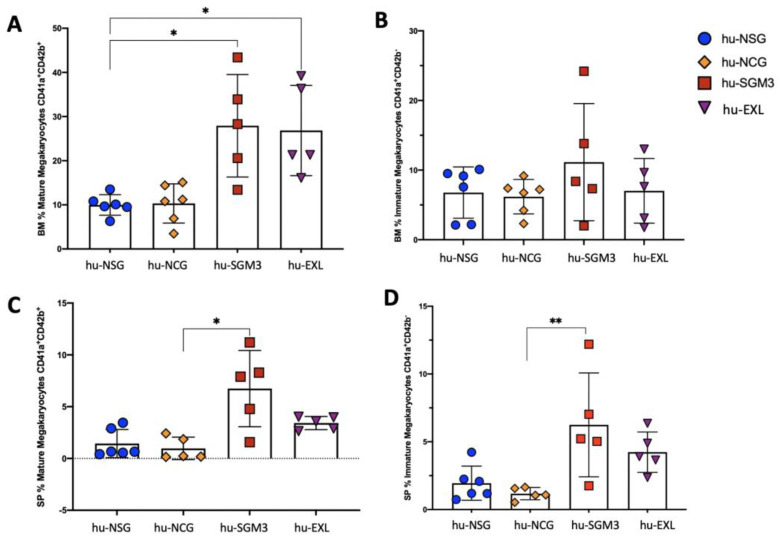
Frequency of human megakaryocytes in tissues of the hu-NSG, hu-NCG, hu-SGM3, and hu-EXL models from weeks 15 and 16 combined. Frequencies of mature megakaryocytes in the bone marrow (**A**), immature megakaryocytes in the bone marrow (**B**), mature megakaryocytes in the spleen (**C**), and immature megakaryocytes in the spleen (**D**). * *p* < 0.005; ** *p* < 0.01 (Dunn test).

**Figure 9 microorganisms-11-01548-f009:**
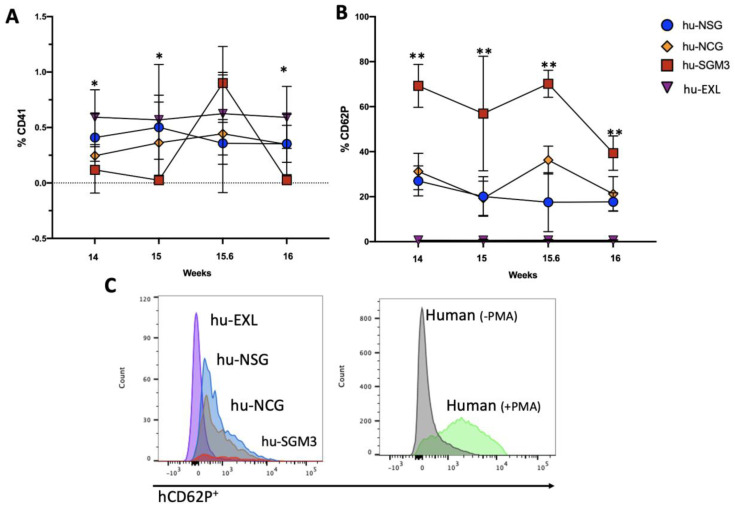
Frequency and activation profile of human platelets in the peripheral blood (PB) of the hu-NSG, hu-NCG, hu-SGM3, and hu-EXL models. Frequency of human CD41^+^ in PB (**A**), platelet activation profile CD41^+^ CD62P^+^ in PB (**B**). Representative graphic of platelet profile activation from humanized mouse models and human samples as positive controls (**C**). PMA refers to platelet monocyte aggregate formation. * Statistical differences between hu-EXL and one or multiple humanized models. ** Statistical differences between hu-SGM3 and one or multiple humanized models (Dunn test).

**Figure 10 microorganisms-11-01548-f010:**
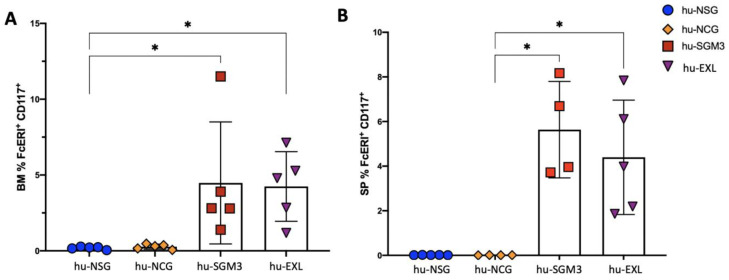
Frequency of human mast cells in tissues of the hu-NSG, hu-NCG, hu-SGM3, and hu-EXL models from weeks 15 and 16 combined. Frequency of FcεRI^+^ CD117^+^ mast cells bone marrow (**A**), immature megakaryocytes in the spleen (**B**). * *p* < 0.005 (Dunn test).

**Figure 11 microorganisms-11-01548-f011:**
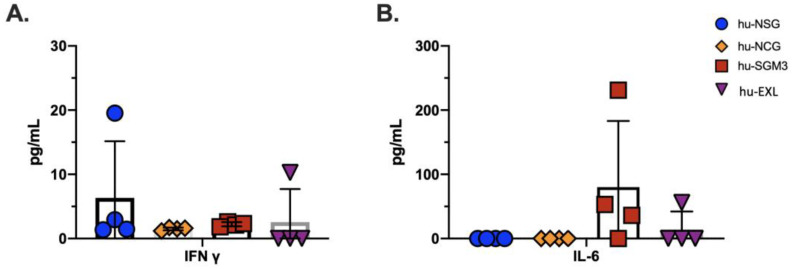
Levels of human cytokines in the peripheral blood of the hu-NSG, hu-NCG, hu-SGM3, and hu-EXL models. (**A**) Levels of human IFN-γ and (**B**) human IL-6. The levels of IFN-α, TNF-α, IL12p70, IL10, and IP10 were undetectable in these models.

**Table 1 microorganisms-11-01548-t001:** The production of human cytokines during human xenograft development.

Cytokine	hu-NSG	hu-NCG	hu-SGM3	hu-EXL
*TNF-α*	−	−	−	−
*IFN-α*	−	−	−	−
*IFN-γ*	+	+	+	+
*IL-6*	−	−	+	+
*IL12p70*	−	−	−	−
*IL10*	−	−	−	−
*IP10*	−	−	−	−

+, Cytokine was detected; −, Cytokine was not detected.

**Table 2 microorganisms-11-01548-t002:** NSG-, NCG-, SGM3-, and EXL-humanized mouse model characteristics and proposed applications in human infectious diseases and cancer.

Model	Advantages	Limitations	Infections Models
hu-NSG	Low cost Short- and long-term engraftment experiments Low proinflammatory environment post-transplant Human immune cell reconstitution **PB:** CD45 lymphocytes, CD4^+^ T cells, CD8^+^ T cells, CD14^+^ monocytes, CD41^+^ platelets **BM:** CD45 lymphocytes, CD4^+^ T cells, CD8^+^ T cells, CD14^+^ monocytes, CD41^+^ megakaryocytes **SP:** CD45 lymphocytes, CD4^+^ T cells, CD8^+^ T cells, CD14^+^ monocytes, CD41^+^ megakaryocytes	Does not promote human hematopoiesis via human cytokines Does not develop mast cells Low numbers of monocytes Low numbers of human platelets GvHD (long-term)	DENV [18], HIV [14,45], Variola virus [46], HBV [47], HCV [48], CMV [49], *P. falciparum* [50]
hu-NCG	Long- and short-term engraftment experiments Low proinflammatory environment post-transplant Human immune cell reconstitution **PB:** CD45 lymphocytes, CD4^+^ T cells, CD8^+^ T cells, CD14^+^ monocytes, CD41^+^ platelets **BM:** CD45 lymphocytes, CD4^+^ T cells, CD8^+^ T cells, CD14^+^ monocytes, CD41^+^ megakaryocytes **SP:** CD45 lymphocytes, CD4^+^ T cells, CD8^+^ T cells, CD14^+^ monocytes, CD41^+^ megakaryocytes	High cost Does not promote human hematopoiesis via human cytokines Does not develop mast cells Low number of monocytes Low number of human platelets GvHD (long-term)	NA
hu-SGM3	Promote human hematopoiesis via human cytokines Best reconstitution of the lymphoid linage Human immune cell reconstitution **PB:** CD45 lymphocytes, CD4^+^ T cells, CD8^+^ T cells, CD14^+^ monocytes, CD41^+^ platelets **BM:** CD45 lymphocytes, CD4^+^ T cells, CD8^+^ T cells, CD14^+^ monocytes, CD41^+^ megakaryocytes, mast cells **SP:** CD45 lymphocytes, CD4^+^ T cells, CD8^+^ T cells, CD14^+^ monocytes, CD41^+^ megakaryocytes, mast cells	High cost Short-term engraftment experiments Hight proinflammatory environment post-transplant Low number and activated profile of human platelets GvHD (in the short-term)	HIV [51], Ebola [52], DENV [53], Staphylococcus aureus [54]
hu-EXL	Promote human hematopoiesis via human cytokines Long and short engraftment experiments Hight proinflammatory environment post-transplant High numbers of human platelets with a rest profile Human immune cell reconstitution **PB:** CD45 lymphocytes, CD4^+^ T cells, CD8^+^ T cells, CD14^+^ monocytes, CD41^+^ platelets **BM:** CD45 lymphocytes, CD4^+^ T cells, CD8^+^ T cells, CD14^+^ monocytes, CD41^+^ megakaryocytes, mast cells **SP:** CD45 lymphocytes, CD4^+^ T cells, CD8^+^ T cells, CD14^+^ monocytes, CD41^+^ megakaryocytes, mast cells	High cost High proinflammatory environment post-transplant GvHD (in the long-term)	HIV [33], *Pneumocystis* [55]

PB, peripheral blood; BM, bone marrow; SP, spleen; DENV, dengue virus; HIV, human immunodeficiency virus; HBV, hepatitis B virus; HCV, hepatitis C virus; CMV, cytomegalovirus; NA, no data available.

## Data Availability

No new data were created or analyzed in this study. Data sharing is not applicable.

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
