# Peer review of "A Comparison of Lymphoid and Myeloid Cells Derived from Human Hematopoietic Stem Cells Xenografted into NOD-Derived Mouse Strains"

_microorganisms, 2023, doi:10.3390/microorganisms11061548_

Round 1

Reviewer 1 Report

The manuscript provides relevant information about the knowing the different animal models available to be used as models to study different human diseases and it`s shows important differences between four strains studied that can influence the interpretation of data when used as a humanized mice model for investigating human cancer, infectious diseases, and graft-versus-host disease.

Major comments:

1. In the Material and Methods section, item: Mice and xenotransplantation (2.2), the authors mentioned that for NSG, NSG-SGM3 and NCG strains the mice were engraftment with 4 to 5 days old while NOG-EXL mice were engraftment  with 14-16 weeks-old.

1.1. Why the engraftment age was different in NOG-EXL strain?

1.2. The engraftment can be influenced by age as mentioned by authors in the discussion "However, comparing successful engraftment between studies is challenging, as it is influenced by various factors such as mouse strain, age, source of CD34+ cells engrafted, preconditioning method and donor variability [15,21-23]". How the authors can justify or explain the difference in engraftment in NOG-EXL strain?

2. In the item Statistical analysis (2.8), the authors mentioned that a nonparametric Wilcoxon rank-sum test was used for data analysis. The data were submitted in a normality test before? If yes, please insert the test name used. If not, please consider review all results if the normality test show a normal distribution.

3. The conclusion is very extensive. Please consider the suggestion to transfer the first two paragraphs for the end of discussion section.

Minor comments:

Line 110: "Plasma samples at day 14 were collected" day 14 refers to animal age or time after engraftment?.

Line 169-170: "In addition to peripheral blood (PB), we established the frequency of those cells in different organs at different time points (Figures 2-4)". The figure 2 show differents time points but the figure 3 and 4 not. Please correct test and add the correct time for figure 3 and 4 analyses.

The figures 6B, 6C, 7F, 8B, 8D, 9C,  were not described in the test. They should be include.

Author Response

Reviewer Comment 1: In the Material and Methods section, item: Mice and xenotransplantation (2.2), the authors mentioned that for NSG, NSG-SGM3 and NCG strains the mice were engraftment with 4 to 5 days old while NOG-EXL mice were engraftment  with 14-16 weeks-old.

Why the engraftment age was different in NOG-EXL strain?

Response: Since NOG-EXL mice are heterozygous for Il2rgtm1Sug gene, and only homozygous Il2rgtm1Sug mice are used for humanization, all pups need to be genotyped before enrolled in any study.  This process takes up to a week to be completed and pups need to be engrafted before day 5. Therefore, there was a technical limitation to use NOG-EXL pups in this experiment. In addition, those commercially available animals were engrafted when there were 14-16 weeks old. We concur that this could be a limitation of this model. A clarification note was added in lines 108-110.

The engraftment can be influenced by age as mentioned by authors in the discussion "However, comparing successful engraftment between studies is challenging, as it is influenced by various factors such as mouse strain, age, source of CD34+ cells engrafted, preconditioning method and donor variability [15,21-23]". How can the authors justify or explain the difference in engraftment in NOG-EXL strain?

Response: The reviewer is correct. However, in the case of NOG-EXL strain, age is one of the factors that could not be controlled due to limitations of its specific genotype as mentioned before. However, we decided to include them in this comparison due to their common use in research.

Reviewer Comment 2: In the item Statistical analysis (2.8), the authors mentioned that a nonparametric Wilcoxon rank-sum test was used for data analysis. The data were submitted in a normality test before? If yes, please insert the test name used. If not, please consider review all results if the normality test show a normal distribution.

Response: Thank you for pointing this out. All the data were submitted to a normality test using a Shapiro-Wilk test. This were clarified in the line 163 and 165 of the document.

Reviewer Comment 3: The conclusion is very extensive. Please consider the suggestion to transfer the first two paragraphs for the end of discussion section.

Response: As suggested by the reviewer, we have summarized the conclusion section and the first paragraph were transfer to the end of discussion section.

Reviewer Comment 4: Line 110: "Plasma samples at day 14 were collected" day 14 refers to animal age or time after engraftment?

Response: Thank you for pointing this out. The following clarification was introduced in Line 113: “Plasma samples at week 14 after xenotransplantation were collected.”

Reviewer Comment 5: Line 169-170: "In addition to peripheral blood (PB), we established the frequency of those cells in different organs at different time points (Figures 2-4)". The figure 2 show different time points but the figure 3 and 4 not. Please correct test and add the correct time for figure 3 and 4 analyses. The figures 6B, 6C, 7F, 8B, 8D, 9C, were not described in the test. They should be included.

Response: Different time points were included in figures 3 and 4. In addition, a description of all the figures was added.

Reviewer 2 Report

1. Methodology is missing in the abstract.

2. Authors need to describe in brief about the individual role of lymphoid and myeloid cells.

3. Construct the relation between cytokines with lymphoid and myeloid cells.

4. Describe the full form of NSG

5. How many mice were used in this study.

6. Describe in detail about flow cytometry analysis.

7. In 2.7 section Among co2, 2 should be subscript

8. Results were well written but authors need to improve the quality of images.

9. Overall, authors can explain what is the importance of this study in the discussion

Author Response

Reviewer Comment 1: Methodology is missing in the abstract.

Response: Thank you for pointing this out. The methodology was added in the abstract Line 17-18.

Reviewer Comment 2: Authors need to describe in brief about the individual role of lymphoid and myeloid cells.

Response: Thank you for this suggestion. We included a brief description of the lymphoid and myeloid cells. Line 342,429,438-439,476-477 and 483-484.

Reviewer Comment 3: Construct the relation between cytokines with lymphoid and myeloid cells.

Response: Thank you for your comment. These relationships are described in Lines: 331-333, 336-338, 429-432, 470-473, 478-482, 503-505, 529-537 and 539-544.

Reviewer Comment 4: Describe the full form of NSG

Response: We have added the suggested content to the manuscript on line 50.

Reviewer Comment 5: How many mice were used in this study.

Answer: As suggested by the reviewer, we have added this information to the text in the lines 95-100 as follow:  “NSG (NOD.Cg-Prkdcscid Il2rgtm1Wjl/SzJ) and NSG-SGM3 (NOD.Cg-Prkdcscid Il2rgtm1Wjl Tg (CMV-IL3,CSF2,KITLG)1Eav/MloySzJ) mice were purchased from Jackson Laboratories (n=12 and n=11). NCG (NOD-Prkdcem26Cd52Il2rgem26Cd22/NjuCrl) mice were purchased from Charles River Laboratories (n=10), and NOG-EXL (NOD.Cg-Prkdcscid Il2rgtm1Sug Tg (SV40/HTLV-IL3,CSF2)10-7Jic/JicTac) mice were purchased from Taconic Biosciences (n=19).

Reviewer Comment 6: Describe in detail about flow cytometry analysis.

Answer: Thank you to the reviewer for his interest in our flow cytometry methodology. In order to make the manuscript easy to read, the gating strategies are included as supplementary material and the readers are referred to our previous publication describing details of this method.

Perdomo-Celis, F.; Medina-Moreno, S.; Davis, H.; Bryant, J.; Zapata, J.C. HIV Replication in Humanized IL-3/GM-CSF-Transgenic NOG Mice. Pathogens 2019, 8, doi:10.3390/pathogens8010033.

Reviewer Comment 7: In 2.7 section Among co2, 2 should be subscript

Response: As suggested by the reviewer. The CO2 in the section 2.7 was described has followed: “Mice were euthanized by CO2 asphyxiation in an acrylic plastic chamber for approximately 6-10 minutes, followed by cervical dislocation…” lines 153-154

Reviewer Comment 8: Results were well written, but authors need to improve the quality of images.

Response: Thank you. The quality of images was improved specially in the supplementary data.

Reviewer Comment 9: Overall, authors can explain what is the importance of this study in the discussion?

Response: Thank you for this suggestion. The importance of the study can be found in lines 548-555 in the discussion section.